# GNN-FiLM: Graph Neural Networks with Feature-wise Linear Modulation

## Abstract

This paper presents a new Graph Neural Network (GNN) type using feature-wise linear modulation (FiLM). Many standard GNN variants propagate information along the edges of a graph by computing "messages" based only on the representation of the source of each edge. In GNN-FiLM, the representation of the target node of an edge is additionally used to compute a transformation that can be applied to all incoming messages, allowing feature-wise modulation of the passed information.

Results of experiments comparing different GNN architectures on three tasks from the literature are presented, based on re-implementations of baseline methods. Hyperparameters for all methods were found using extensive search, yielding somewhat surprising results: differences between baseline models are smaller than reported in the literature. Nonetheless, GNN-FiLM outperforms baseline methods on a regression task on molecular graphs and performs competitively on other tasks.

## 1 Introduction

Learning from graph-structured data has seen explosive growth over the last few years, as graphs are a convenient formalism to model the broad class of data that has objects (treated as vertices) with some known relationships (treated as edges). Example usages include reasoning about physical and biological systems, knowledge bases, computer programs, and relational reasoning in computer vision tasks. This graph construction is a highly complex form of feature engineering, mapping the knowledge of a domain expert into a graph structure which can be consumed and exploited by high-capacity neural network models.

Many neural graph learning methods can be summarised as neural message passing (Gilmer et al., 2017): nodes are initialised with some representation and then exchange information by transforming their current state (in practice with a single linear layer) and sending it as a message to all neighbours in the graph. At each node, messages are aggregated in some way and then used to update the associated node representation. In this setting, the message is entirely determined by the source node (and potentially the edge type) and the target node is not taken into consideration. A (partial) exception to this is the family of Graph Attention Networks (Veličković et al., 2018), where the agreement between source and target representation of an edge is used to determine the *weight* of the message in an attention architecture. However, this weight is applied to all dimensions of the message at the same time.

A simple consequence of this observation may be to simply compute messages from the pair of source and target node state. However, the linear layer commonly used to compute messages would only allow additive interactions between the representations of source and target nodes. More complex transformation functions are often impractical, as computation in GNN implementations is dominated by the message transformation function.

However, this need for non-trivial interaction between different information sources is a common problem in neural network design. A recent trend has been the use of *hypernetworks* (Ha et al., 2017), neural networks that compute the weights of other networks. In this setting, interaction between two signal sources is achieved by using one of them as the input to a hypernetwork and the other as input to the computed network. While an intellectually pleasing approach, it is often impractical because the prediction of weights of non-trivial neural networks is computationally expensive.

Approaches to mitigate this exist (e.g., Wu et al. (2019) handle this in natural language processing), but are often domain-specific.

A more general mitigation method is to restrict the structure of the computed network. Recently, "feature-wise linear modulations" (FiLM) were introduced in the visual question answering domain (Perez et al., 2017). Here, the hypernetwork is fed with an encoding of a question and produces an element-wise affine function that is applied to the features extracted from a picture. This can be adapted to the graph message passing domain by using the representation of the target node to compute the affine function. This compromise between expressiveness and computational feasibility has been very effective in some domains and the results presented in this article indicate that it is also a good fit for the graph domain.

This article explores the use of hypernetworks in learning on graphs. Sect. 2 first reviews existing GNN models from the related work to identify commonalities and differences. This involves generalising a number of existing formalisms to new formulations that are able to handle graphs with different types of edges, which are often used to model different relationship between vertices. Then, two new formalisms are introduced: Relational Graph Dynamic Convolutional Networks (RGDCN), which dynamically compute the neural message passing function as a linear layer, and Graph Neural Networks with Feature-wise Linear Modulation (GNN-FiLM), which combine learned message passing functions with dynamically computed element-wise affine transformations. In Sect. 3, a range of baselines are compared in extensive experiments on three tasks from the literature, spanning classification, regression and ranking tasks on small and large graphs. Experiments were performed on re-implementations of existing model architectures in the same framework and hyperparameter setting searches were performed with the same computational budgets across all architectures. The results show that differences between baselines are smaller than the literature suggests and that the new FiLM model performs well on a number of interesting tasks.

## 2 MODEL

**Notation.** Let $\mathcal{L}$ be a finite (usually small) set of edge types. Then, a directed graph $\mathcal{G} = (\mathcal{V}, \mathcal{E})$ has nodes $\mathcal{V}$ and typed edges $\mathcal{E} \subseteq \mathcal{V} \times \mathcal{L} \times \mathcal{V}$, where $(u, \ell, v) \in \mathcal{E}$ denotes an edge from node $u$ to node $v$ of type $\ell$, usually written as $u \xrightarrow{\ell} v$.

**Graph Neural Networks.** As discussed above, Graph Neural Networks operate by propagating information along the edges of a given graph. Concretely, each node $v$ is associated with an initial representation $\boldsymbol{h}_v^{(0)}$ (for example obtained from the label of that node, or by some other model component). Then, a GNN layer updates the node representations using the node representations of its neighbours in the graph, yielding representations $\boldsymbol{h}_v^{(1)}$. This process can be unrolled through time by repeatedly applying the same update function, yielding representations $\boldsymbol{h}_v^{(2)} \ldots \boldsymbol{h}_v^{(T)}$. Alternatively, several GNN layers can be stacked, which is intuitively similar to unrolling through time, but increases the GNN capacity by using different parameters for each timestep.

In Gated Graph Neural Networks (GGNN) (Li et al., 2016), the update rule uses one linear layer $\boldsymbol{W}_\ell$ per edge type $\ell$ to compute messages and combines the aggregated messages with the current representation of a node using a recurrent unit $r$ (e.g., GRU or LSTM cells), yielding the following definition.

$$\boldsymbol{h}_v^{(t+1)} = r\left(\boldsymbol{h}_v^{(t)}, \sum_{u \xrightarrow{\ell} v \in \mathcal{E}} \boldsymbol{W}_\ell \boldsymbol{h}_u^{(t)} \; ; \; \boldsymbol{\theta}_r\right) \tag{1}$$

The learnable parameters of the model are the edge-type-dependent weights $\boldsymbol{W}_\ell$ and the recurrent cell parameters $\boldsymbol{\theta}_r$.

In Relational Graph Convolutional Networks (R-GCN) (Schlichtkrull et al., 2018), the gated unit is replaced by a simple non-linearity $\sigma$ (e.g., the hyperbolic tangent).

$$\boldsymbol{h}_v^{(t+1)} = \sigma\left(\sum_{u \xrightarrow{\ell} v \in \mathcal{E}} \frac{1}{c_{v,\ell}} \cdot \boldsymbol{W}_\ell \boldsymbol{h}_u^{(t)}\right) \tag{2}$$

Here, $c_{v,\ell}$ is a normalisation factor usually set to the number of edges of type $\ell$ ending in $v$. The learnable parameters of the model are the edge-type-dependent weights $\boldsymbol{W}_\ell$. It is important to note

that in this setting, the edge type set $\mathcal{L}$ is assumed to contain a special edge type 0 for self-loops $v \xrightarrow{0} v$, allowing state associated with a node to be kept.

In Graph Attention Networks (GAT) (Veličković et al., 2018), new node representations are computed from a weighted sum of neighbouring node representations. The model can be generalised from the original definitional to support different edge types as follows (we will call this R-GAT below).[1]

$$
\begin{aligned}
e_{u,\ell,v} &= \text{LeakyReLU}(\boldsymbol{\alpha}_\ell \cdot (\boldsymbol{W}_\ell \boldsymbol{h}_u^{(t)} \| \boldsymbol{W}_\ell \boldsymbol{h}_v^{(t)})) \\
\boldsymbol{a}_v &= \text{softmax}(e_{u,\ell,v} \mid u \xrightarrow{\ell} v \in \mathcal{E})
\end{aligned}
$$

$$
\boldsymbol{h}_v^{(t+1)} = \sigma \left( \sum_{u \xrightarrow{\ell} v \in \mathcal{E}} (\boldsymbol{a}_v)_{u \xrightarrow{\ell} v} \cdot \boldsymbol{W}_\ell \boldsymbol{h}_u^{(t)} \right) \tag{3}
$$

Here, $\boldsymbol{\alpha}_\ell$ is a learnable row vector used to weigh different feature dimensions in the computation of an attention ("relevance") score of the node representations, $\boldsymbol{x} \| \boldsymbol{y}$ is the concatenation of vectors $\boldsymbol{x}$ and $\boldsymbol{y}$, and $(\boldsymbol{a}_v)_{u \xrightarrow{\ell} v}$ refers to the weight computed by the softmax for that edge. The learnable parameters of the model are the edge-type-dependent weights $\boldsymbol{W}_\ell$ and the attention parameters $\boldsymbol{\alpha}_\ell$. In practice, GATs usually employ several attention heads that independently implement the mechanism above in parallel, using separate learnable parameters. The results of the different attention heads are then concatenated after each propagation round to yield the value of $\boldsymbol{h}_v^{(t+1)}$.

More recently, Xu et al. (2019) analysed the expressiveness of different GNN types, comparing their ability to distinguish similar graphs with the Weisfeiler-Lehman (WL) graph isomorphism test. Their results show that GCNs and the GraphSAGE model Hamilton et al. (2017) are strictly weaker than the WL test and hence they developed Graph Isomorphism Networks (GIN) (Xu et al., 2019), which are indeed as powerful as the WL test. While the GIN definition is limited to a single edge type, Corollary 6 of Xu et al. (2019) shows that using the definition

$$
\boldsymbol{h}_v^{(t+1)} = \varphi \left( (1 + \epsilon) \cdot f(\boldsymbol{h}_v^{(t)}) + \sum_{u \to v \in \mathcal{E}} f(\boldsymbol{h}_u^{(t)}) \right),
$$

there are choices for $\epsilon$, $\varphi$ and $f$ such that the node representation update is sufficient for the overall network to be as powerful as the WL test. In the setting of different edge types, the function $f$ in the sum over neighbouring nodes needs to reflect different edge types to distinguish graphs such as $v \xrightarrow{1} u \xleftarrow{2} w$ and $v \xrightarrow{2} u \xleftarrow{1} w$ from each other. Using different functions $f_\ell$ for different edge types makes it possible to unify the use of the current node representation $\boldsymbol{h}_v^{(t)}$ with the use of neighbouring node representations by again using a fresh edge type 0 for self-loops $v \xrightarrow{0} v$. In that setting, the factor $(1 + \epsilon)$ can be integrated into $f_0$. Finally, following an argument similar to Xu et al. (2019), $\varphi$ and $f$ at subsequent layers can be "merged" into a single function which can be approximated by a multilayer perceptron (MLP), yielding the final R-GIN definition

$$
\boldsymbol{h}_v^{(t+1)} = \sigma \left( \sum_{u \xrightarrow{\ell} v \in \mathcal{E}} MLP(\boldsymbol{h}_u^{(t)} \, ; \, \boldsymbol{\theta}_\ell) \right). \tag{4}
$$

The learnable parameters here are the edge-specific weights $\boldsymbol{\theta}_\ell$. Note that Eq. (4) is very similar to the definition of R-GCNs (Eq. (2)), only dropping the normalisation factor $\frac{1}{c_{v,\ell}}$ and replacing linear layers by an MLP.

While many more GNN variants exist, the four formalisms above are broadly representative of general trends. It is notable that in all of these models, the information passed from one node to another is based on the learned weights and the representation of the source of an edge. In contrast, the representation of the *target* of an edge is only updated (in the GGNN case Eq. (1)), treated as another incoming message (in the R-GCN case Eq. (2) and the R-GIN case Eq. (4)), or used to weight the relevance of an edge (in the R-GAT case Eq. (3)). Sometimes unnamed GNN variants of the above are used (e.g., by Selsam et al. (2019); Paliwal et al. (2019)), replacing the linear layers to compute the messages for each edge by MLPs applied to the concatenation of the representations

---

[1]Note that this is similar to the ARGAT model presented by Busbridge et al. (2019), but unlike the models studied there (and like the original GATs) uses a single linear layer to compute attention scores $e_{u,\ell,v}$, instead of simpler additive or multiplicative variants.

of source and target nodes. In the experiments, this will be called GNN-MLP, formally defined as follows.[2]

$$h_v^{(t+1)} = \sigma \left( \sum_{u \xrightarrow{\ell} v \in \mathcal{E}} \frac{1}{c_{v,\ell}} \cdot MLP\left( h_u^{(t)} \| h_v^{(t)} \; ; \; \boldsymbol{\theta}_\ell \right) \right) \tag{5}$$

Below, we will instantiate the $MLP$ with a single linear layer to obtain what we call GNN-MLP0, which only differs from R-GCNs (Eq. (2)) in that the message passing function is applied to the concatenation of source and target state.

## 2.1 GRAPH HYPERNETWORKS

Hypernetworks (i.e., neural networks computing the parameters of another neural network) (Ha et al., 2017) have been successfully applied to a number of different tasks; naturally raising the question if they are also applicable in the graph domain.

Intuitively, a hypernetwork corresponds to a higher-order function, i.e., it can be viewed as a function computing another function. Hence, a natural idea would be to use the target of a message propagation step to compute the function computing the message; essentially allowing it to focus on features that are especially relevant for the update of the target node representation.

**Relational Graph Dynamic Convolutional Networks (RGDCN)**  A first attempt would be to adapt (2) to replace the learnable message transformation $W_\ell$ by the result of some learnable function $f$ that operates on the target representation:

$$h_v^{(t+1)} = \sigma \left( \sum_{u \xrightarrow{\ell} v \in \mathcal{E}} f(h_v^{(t)} \; ; \; \boldsymbol{\theta}_{f,\ell}) h_u^{(t)} \right)$$

However, for a representation size $D$, $f$ would need to produce a matrix of size $D^2$ from $D$ inputs. Hence, if implemented as a simple linear layer, $f$ would have on the order of $\mathcal{O}(D^3)$ parameters, quickly making it impractical in most contexts.

This can be somewhat mitigated by splitting the node representations $h_v^{(t)}$ into $C$ "chunks" $h_{v,c}^{(t)}$ of dimension $K = \frac{D}{C}$:

$$W_{\ell,t,v,c} = f(h_v^{(t)} \; ; \; \boldsymbol{\theta}_{f,\ell,c})$$

$$h_v^{(t+1)} = \underset{1 \leq c \leq C}{\|} \sigma \left( \sum_{u \xrightarrow{\ell} v \in \mathcal{E}} W_{\ell,t,v,c} h_{u,c}^{(t)} \right) \tag{6}$$

The number of parameters of the model can now be reduced by tying the value of some instances of $\boldsymbol{\theta}_{f,\ell,c}$. For example, the update function for a chunk $c$ can be computed using only the corresponding chunk of the node representation $h_{v,c}^{(t)}$, or the same update function can be applied to all "chunks" by setting $\boldsymbol{\theta}_{f,\ell,1} = \ldots = \boldsymbol{\theta}_{f,\ell,C}$. The learnable parameters of the model are only the hypernetwork parameters $\boldsymbol{\theta}_{f,\ell,c}$. This is somewhat less desirable than the related idea of Wu et al. (2019), which operates on sequences, where sharing between neighbouring elements of the sequence has an intuitive interpretation that is not applicable in the general graph setting.

**Graph Neural Networks with Feature-wise Linear Modulation (GNN-FiLM)**  In (6), the message passing layer is a linear transformation conditioned on the target node representation, focusing on separate chunks of the node representation at a time. In the extreme case in which the dimension of each chunk is 1, this method coincides with the ideas of Perez et al. (2017), who propose to use layers of element-wise affine transformations to modulate feature maps in the visual question answering setting; there, a natural language question is the input used to compute the affine transformation applied to the features extracted from a picture.

In the graph setting, we can use each node's representation as an input that determines an element-wise affine transformation of incoming messages, allowing the model to dynamically up-weight

---

[2] These are similar to R-GIN, but apply an MLP to the concatenation of source and target state for each message.

and down-weight features based on the information present at the target node of an edge. This yields the following update rule, using a learnable function $g$ to compute the parameters of the affine transformation.

$$\boldsymbol{\beta}_{\ell,v}^{(t)}, \boldsymbol{\gamma}_{\ell,v}^{(t)} = g(\boldsymbol{h}_v^{(t)} \; ; \; \boldsymbol{\theta}_{g,\ell})$$

$$\boldsymbol{h}_v^{(t+1)} = \sigma \left( \sum_{u \xrightarrow{\ell} v \in \mathcal{E}} \boldsymbol{\gamma}_{\ell,v}^{(t)} \odot \boldsymbol{W}_\ell \boldsymbol{h}_u^{(t)} + \boldsymbol{\beta}_{\ell,v}^{(t)} \right) \qquad (7)$$

The learnable parameters of the model are both the hypernetwork parameters $\boldsymbol{\theta}_{g,\ell}$ and the weights $\boldsymbol{W}_\ell$. In practice, implementing $g$ as a single linear layer works well.

In the case of using a single linear layer, the resulting message passing function is bilinear in source and target node representation, as the message computation is centred around $(\boldsymbol{W}_g \boldsymbol{h}_v^{(t)}) \odot (\boldsymbol{W}_\ell \boldsymbol{h}_u^{(t)})$. This is the core difference to the (linear) interaction of source and target node representations in models that use $\boldsymbol{W}_\ell (\boldsymbol{h}_u^{(t)} \| \boldsymbol{h}_v^{(t)})$.

A simple toy example may illustrate the usefulness of such a mechanism: assuming a graph of nodes $\mathcal{V}_A$ and $\mathcal{V}_B$ and edge types 1 and 2, a task may involve counting the number of 1-neighbours of $\mathcal{V}_A$ nodes and of 2-neighbours of $\mathcal{V}_B$ nodes. By setting $\gamma_{1,v_a} = 1$, $\gamma_{2,v_a} = 0$ for $v_a \in \mathcal{V}_A$ and $\gamma_{1,v_b} = 0$, $\gamma_{2,v_b} = 1$ for $v_b \in \mathcal{V}_B$, GNN-FiLM can solve this in a single layer. Simpler approaches can solve this by counting $A/1$, $A/2$, $B/1$ and $B/2$ neighbours separately in one layer and then projecting to the correct counter, but require more feature dimensions and layers for this. As this toy example illustrates, a core capability of GNN-FiLM is to learn to ignore graph edges based on the representation of target nodes.

Note that the featurewise modulation can also be viewed of an extension of the gating mechanism of GRU or LSTM cells used in GGNNs. Concretely, the "forgetting" of memories in a GRU/LSTM is similar to down-weighting messages computed for the self-loop edges and the gating of the cell input is similar to the modulation of other incoming messages. However, GGNNs apply this gating to the sum of all incoming messages (cf. Eq. (1), wheras in GNN-FiLM the modulation additionally depends on the edge type, allowing for a more fine-grained gating mechanism.

Finally, a small implementation bug brought focus to the fact that applying the non-linearity $\sigma$ after summing up messages from neighbouring nodes can make it harder to perform tasks such as counting the number of neighbours with a certain feature. In experiments, applying the non-linearity before aggregation as in the following update rule improved performance.

$$\boldsymbol{h}_v^{(t+1)} = l \left( \sum_{u \xrightarrow{\ell} v \in \mathcal{E}} \sigma \left( \boldsymbol{\gamma}_{\ell,v}^{(t)} \odot \boldsymbol{W}_\ell \boldsymbol{h}_u^{(t)} + \boldsymbol{\beta}_{\ell,v}^{(t)} \right) \; ; \; \boldsymbol{\theta}_l \right) \qquad (8)$$

However, this means that the magnitude of node representations is now dependent on the degree of nodes in the handled graph. This can sometimes lead to instability during training, which can in turn be controlled by adding an additional layer $l$ after message passing, which can be a simple bounded nonlinearity (e.g. $tanh$), a fully connected layer, or layer normalisation (Ba et al., 2016), or any combination of these.

## 3 EVALUATION

### 3.1 GNN BENCHMARK TASKS

Due to the versatile nature of the GNN modelling formalism, many fundamentally different tasks are studied in the research area and it should be noted that good results on one task often do not transfer over to other tasks. This is due to the widely varying requirements of different tasks, as the following summary of tasks from the literature should illustrate.

- Cora/Citeseer/Pubmed (Sen et al., 2008): Each task consists of a single graph of $\sim 10\,000$ nodes corresponding to documents and undirected (sic!) edges corresponding to references. The sparse $\sim 1\,000$ node features are a bag of words representation of the corresponding documents. The goal is to assign a subset of nodes to a small number of classes. State of the art performance on these tasks is achieved with two propagation steps along graph edges.

- PPI (Zitnik & Leskovec, 2017): A protein-protein interaction dataset consisting of 24 graphs of $\sim 2\,500$ nodes corresponding to different human tissues. Each node has 50 features selected by domain experts and the goal is node-level classification, where each node may belong to several of the 121 classes. State of the art performance on this task requires three propagation steps.

- QM9 property prediction (Ramakrishnan et al., 2014): $\sim 130\,000$ graphs of $\sim 8$ nodes represent molecules, where nodes are heavy atoms and undirected, typed edges are bonds between these atoms, different edge types indicating single/double/etc. bonds. The goal is to regress each graph to a number of quantum chemical properties. State of the art performance on these tasks requires at least four propagation steps.

- VarMisuse (Allamanis et al., 2018): $\sim 235\,000$ graphs of $\sim 2\,500$ nodes each represent program fragments, where nodes are tokens in the program text and different edge types represent the program's abstract syntax tree, data flow between variables, etc. The goal is to select one of a set of candidate nodes per graph. State of the art performance requires at least six propagation steps.

Hence, tasks differ in the complexity of edges (from undirected and untyped to directed and many-typed), the size of the considered graphs, the size of the dataset, the importance of node-level vs. graph-level representations, and the number of required propagation steps.

This article includes results on the PPI, QM9 and VarMisuse tasks. Preliminary experiments on the citation network data showed results that were at best comparable to the baseline methods, but changes of a random seed led to substantial fluctuations (mirroring the problems with evaluation on these tasks reported by Shchur et al. (2018)).

## 3.2 Implementation

To allow for a wider comparison, the implementation of GNN-FiLM is accompanied by implementations of a range of baseline methods. These include GGNN (Li et al., 2016) (see Eq. (1)), R-GCN (Schlichtkrull et al., 2018) (see Eq. (2)), R-GAT (Veličković et al., 2018) (see Eq. (3)), and R-GIN (Hamilton et al., 2017) (see Eq. (4))[3]. Additionally, GNN-MLP0 is a variant of R-GCN using a single linear layer to compute the edge message from both source and target state (i.e., Eq. (5) instantiated with an "MLP" without hidden layers), and GNN-MLP1 is the same with a single hidden layer. The baseline methods were re-implemented in TensorFlow and individually tested to reach performance equivalent to results reported in their respective source papers. All code for the implementation of these GNNs is released on `https://revealed/after/double/blind/lifted`, together with implementations of all tasks and scripts necessary to reproduce the results reported in this paper. This includes the hyperparameter settings found by search, which are stored in `tasks/default_hypers/` and are selected by default on the respective tasks. The code is designed to facilitate testing new GNN types on existing tasks and easily adding new tasks, allowing for rapid evaluation of new architectures.

Early on in the experiments, it became clear that the RGDCN approach (Eq. (6)) as presented is infeasible. It is extremely sensitive to the parameter initialisation and hence changes to the random seed lead to wild swings in the target metrics. Hence, no experimental results are reported for it in the following. It is nonetheless included in the article (and the implementation) to show the thought process leading to GNN-FiLM, as well as to allow other researchers to build upon this. In the following, GNN-FiLM refers to the formulation of Eq. (8), which performed better than the variant of Eq. (7) across all experiments. Somewhat surprisingly, the same trick (of moving the non-linearity before the message aggregation step) did not help the other GNN types. For all models, using each layer only for a single propagation step performed better than using fewer layers with several propagation steps.

In all experiments, models were trained until the target metric did not improve anymore for some additional epochs (25 for PPI and QM9, 5 for VarMisuse). The reported results on the held-out test data are averaged across the results of a number of training runs, each starting from different random parameter initializations.

---

[3]Note that Eq. (3) and Eq. (4) define generalisations to different edge types not present in the original papers.

### 3.3 EXPERIMENTAL RESULTS

#### 3.3.1 PROTEIN-PROTEIN INTERACTIONS (PPI)

The models are first evaluated on the node-level classification PPI task (Zitnik & Leskovec, 2017), following the dataset split from earlier papers. Training hence used a set of 20 graphs and validation and test sets of two separate graphs each. The graphs use two edge types: the dataset-provided un-typed edges as well as a fresh "self-loop" edge type to allows nodes to keep state across propagation steps.

Hyperparameters for all models were selected based on results from earlier papers and a small grid search of a number of author-selected hyperparameter ranges (see App. A for details). This resulted in three (R-GAT), four (GGNN, GNN-FiLM, GNN-MLP1, R-GCN), or five (GNN-MLP0, R-GIN) layers (propagation steps) and a node representation size of 256 (GNN-MLP0, R-GIN) or 320 (all others). All models use dropout on the node representations before all GNN layers, with a keep ratio of 0.9. After selecting hyperparameters, all models were trained ten times with different random seeds on a NVidia V100.

Tab. 1 shows the micro-averaged F1 score on the classification task on the test graphs, with standard deviations and training times in seconds computed over the ten runs. The results for all re-implemented models are better than the results reported by Veličković et al. (2018) for the GAT model (without edge types). A cursory exploration of the reasons yielded three factors. First, the generalisation to different edge types (cf. Eq. (3)) and the subsequent use of a special self-loop edge type helps R-GAT (and all other models) significantly. Second, using dropout between layers significantly im-

Table 1: GNN results on PPI task. GAT* result taken from Veličković et al. (2018).

| Model | Avg. Micro-F1 | Time (s) |
|---|---|---|
| GAT* | 0.973 ±0.002 | n/a |
| GGNN | 0.990 ±0.001 | 432.6 |
| R-GCN | 0.989 ±0.000 | 759.0 |
| R-GAT | 0.989 ±0.001 | 782.3 |
| R-GIN | 0.991 ±0.001 | 704.8 |
| GNN-MLP0 | **0.992**±0.000 | 556.9 |
| GNN-MLP1 | **0.992**±0.001 | 479.2 |
| GNN-FiLM | **0.992**±0.000 | 308.1 |

proved the results. Third, the larger node representation sizes (compared to 256 used by Veličković et al. (2018)) improved the results again. However, the new GNN-FiLM improves slightly over these four baselines from the literature, while converging substantially faster than all baselines, mainly because it converges in significantly fewer training steps (approx. 240 epochs compared to 400-600 epochs for the other models).

#### 3.3.2 QUANTUM CHEMISTRY (QM9)

All models were additionally evaluated on graph-level regression tasks on the QM9 molecule data set (Ramakrishnan et al., 2014), considering thirteen different quantum chemical properties. The $\sim130k$ molecular graphs in the dataset were split into training, validation and test data by randomly selecting 10 000 graphs for the latter two sets. Additionally, another data split without a test set was used for the hyperparameter search (see below). The graphs use five edge types: the dataset-provided typed edges (single, double, triple and aromatic bonds between atoms) as well as a fresh "self-loop" edge type that allows nodes to keep state across propagation steps. The evaluation differs from the setting reported by Gilmer et al. (2017), as no additional molecular information is encoded as edge features, nor are the graphs augmented by master nodes or additional edges.[4]

Hyperparameters for all models were found using a staged search process. First, 500 hyperparameter configurations were sampled from an author-provided search space (see App. A for details) and run on the first three regression tasks. The top three configurations for each of these three tasks were then run on all thirteen tasks and the final configuration was chosen as the one with the lowest average mean absolute error across all properties, as evaluated on the validation data of that dataset split. This process led to eight layers / propagation steps for all models but GGNN and R-GIN, which showed best performance with six layers. Furthermore, all models used residual connections connecting every second layer and GGNN, R-GCN, GNN-FiLM and GNN-MLP0 additionally used layer normalisation (as in Eq. (8)).

---

[4]Adding these features is straightforward, but orthogonal to the comparison of different GNN variants.

Table 2: GNN average error rates and standard deviations on QM9 target values.

| Property | GGNN | R-GCN | R-GAT | R-GIN | GNN-MLP0 | GNN-MLP1 | GNN-FiLM |
|----------|------|-------|-------|-------|----------|----------|----------|
| mu | 3.85 ±0.16 | 3.21 ±0.06 | 2.68 ±0.06 | 2.64 ±0.11 | **2.36** ±0.04 | 2.44 ±0.12 | 2.38 ±0.13 |
| alpha | 5.22 ±0.86 | 4.22 ±0.45 | 4.65 ±0.44 | 4.67 ±0.52 | 4.27 ±0.36 | 4.63 ±0.54 | **3.75** ±0.11 |
| HOMO | 1.67 ±0.07 | 1.45 ±0.01 | 1.48 ±0.03 | 1.42 ±0.01 | 1.25 ±0.04 | 1.29 ±0.06 | **1.22** ±0.07 |
| LUMO | 1.74 ±0.06 | 1.62 ±0.04 | 1.53 ±0.07 | 1.50 ±0.09 | 1.35 ±0.04 | 1.50 ±0.19 | **1.30** ±0.05 |
| gap | 2.60 ±0.06 | 2.42 ±0.14 | 2.31 ±0.06 | 2.27 ±0.09 | 2.04 ±0.05 | 2.06 ±0.10 | **1.96** ±0.06 |
| R2 | 35.94 ±35.68 | 16.38 ±0.49 | 52.39 ±42.58 | 15.63 ±1.40 | **14.86** ±1.62 | 15.81 ±1.42 | 15.59 ±1.38 |
| ZPVE | 17.84 ±3.61 | 17.40 ±3.56 | 14.87 ±2.88 | 12.93 ±1.81 | 12.00 ±1.66 | 14.12 ±1.10 | **11.00** ±0.74 |
| U0 | 8.65 ±2.46 | 7.82 ±0.80 | 7.61 ±0.46 | 5.88 ±1.01 | 5.55 ±0.38 | 6.94 ±0.64 | **5.43** ±0.96 |
| U | 9.24 ±2.26 | 8.24 ±1.25 | 6.86 ±0.53 | 18.71 ±23.36 | 6.20 ±0.88 | 7.00 ±1.06 | **5.95** ±0.46 |
| H | 9.35 ±0.96 | 9.05 ±1.21 | 7.64 ±0.92 | 5.62 ±0.81 | 5.96 ±0.45 | 7.98 ±0.88 | **5.59** ±0.57 |
| G | 7.14 ±1.15 | 7.00 ±1.51 | 6.54 ±0.36 | 5.38 ±0.75 | **5.09** ±0.57 | 7.14 ±0.51 | 5.17 ±1.13 |
| Cv | 8.86 ±9.07 | 3.93 ±0.48 | 4.11 ±0.27 | 3.53 ±0.37 | **3.38** ±0.20 | 4.60 ±0.74 | 3.46 ±0.21 |
| Omega | 1.57 ±0.53 | 1.02 ±0.05 | 1.48 ±0.87 | 1.05 ±0.11 | **0.84** ±0.02 | 5.60 ±8.82 | 0.98 ±0.06 |

Table 3: Accuracy on VarMisuse task. GGNN[*] result taken from appendix of Allamanis et al. (2018).

| Model | TRAIN | VALID | SEENPROJTEST | UNSEENPROJTEST |
|-------|-------|-------|--------------|----------------|
| GGNN[*] | n/a | n/a | 84.0   n/a | 74.1   n/a |
| GGNN | 87.5 ±1.8% | 82.1 ±0.9% | 85.7 ±0.5% | 79.3 ±1.2% |
| R-GCN | 88.7 ±3.1% | 85.7 ±1.6% | **87.2** ±1.5% | **81.4** ±2.3% |
| R-GAT | 90.4 ±3.9% | 84.2 ±1.0% | 86.9 ±0.7% | 81.2 ±0.9% |
| R-GIN | 93.4 ±1.8% | 84.2 ±1.0% | 87.1 ±0.1% | 81.1 ±0.9% |
| GNN-MLP0 | 95.3 ±2.4% | 83.4 ±0.3% | 86.5 ±0.2% | 80.5 ±1.4% |
| GNN-MLP1 | 94.7 ±1.2% | 84.4 ±0.4% | 86.9 ±0.3% | **81.4** ±0.7% |
| GNN-FiLM | 94.3 ±1.0% | 84.6 ±0.6% | 87.0 ±0.2% | 81.3 ±0.9% |

Each model was trained for each of the properties separately five times using different random seeds on compute nodes with NVidia P100 cards. The average results of the five runs are reported in Tab. 2, with their respective standard deviations.[5] The results indicate that the new GNN-FiLM model outperforms the standard baselines on all tasks and the usually not considered GNN-MLP variants on the majority of tasks.

### 3.3.3 VARIABLE USAGE IN PROGRAMS (VARMISUSE)

Finally, the models were evaluated on the VarMisuse task of Allamanis et al. (2018). This task requires to process a graph representing an abstraction of a program fragment and then select one of a few candidate nodes (representing program variables) based on the representation of another node (representing the location to use a variable in). The experiments are performed using the released split of the dataset, which contains $\sim 130k$ training graphs, $\sim 20k$ validation graphs and two test sets: SEENPROJTEST, which contains $\sim 55k$ graphs extracted from open source projects that also contributed data to the training and validation sets, and UNSEENPROJTEST, which contains $\sim 30k$ graphs extracted from completely unseen projects.

Due to the inherent cost of training models on this dataset (Balog et al. (2019) provide an in-depth performance analysis), a limited hyperparameter grid search was performed, with only $\sim 30$ candidate configurations for each model (see App. A for details). For each model, the configuration yielding the best results on the validation data set fold was selected. This led to six layers for GGNN and R-GIN, eight layers for R-GAT and GNN-MLP0, and ten layers for the remaining models. Graph node hidden sizes were 128 for all models but GGNN and R-GAT, which performed better with 96 dimensions.

The results, shown in Tab. 3, are somewhat surprising, as they indicate a different ranking of model architectures as the results on PPI and QM9, with R-GCN performing best. All re-implemented

---

[5]Note that training sometimes did not converge (as visible in the large standard deviation). Removing these outliers, GGNN achieved 18.11(±1.62) and R-GAT achieved 17.66(±1.23) on R2; R-GIN has an average error rate of 7.04(±1.41) on U, and GNN-MLP1's result on the Omega task is 1.19(±0.08).

baselines beat the results reported by Allamanis et al. (2018), who also reported that R-GCN and GGNN show very similar performance. This is in spite of a simpler implementation of the task than in the original paper, as it only uses the string labels of nodes for the representation and does not use the additional type information provided in the dataset. However, the re-implementation of the task uses the insights from Cvitkovic et al. (2019), who use character CNNs to encode node labels and furthermore introduce extra nodes for subtokens appearing in labels of different nodes, connecting them to their sources (e.g., nodes labelled `openWullfrax` and `closeWullfrax` are both connected to a fresh `Wullfrax` node).

A deeper investigation results showed that the more complex models seem to suffer from significant overfitting to the training data, as can be seen in the results for training and validation accuracy reported in Tab. 3. A brief exploration of more aggressive regularisation methods (more dropout, weight decay) showed no improvement and a deeper understanding of the cause of these results remains for future work.

Furthermore, the large variance in results on the validation set (especially for R-GCN) makes it likely that the hyperparameter grid search with only one training run per configuration did not yield the best configuration for each model.

## 4 DISCUSSION & CONCLUSIONS

After a review of existing graph neural network architectures, the idea of using hypernetwork-inspired models in the graph setting was explored. This led to two models, Graph Dynamic Convolutional Networks and GNNs with feature-wise linear modulation, were presented. While GDCNs seem to be impractical to train, experiments show that GNN-FiLM is competitive with or improving on baseline models on three tasks from the literature.

The extensive experiments also show that a number of results from the literature could benefit from more substantial hyperparameter search and are often missing comparisons to a number of obvious baselines:

- The results in Tab. 1 indicate that GATs have no advantage over GGNNs or R-GCNs on the PPI task, which does not match the findings by Veličković et al. (2018).

- The results in Tab. 3 indicate that R-GCNs are outperforming GGNNs substantially on the VarMisuse task, contradicting the findings of Allamanis et al. (2018).

- The GNN-MLP models are obvious extensions that are often alluded to, but are not part of the usually considered set of baseline models. Nonetheless, experiments across all three tasks have shown that these methods outperform better-published techniques such as GGNNs, R-GCNs and GATs, without a substantial runtime penalty.

These results indicate that there is substantial value in independent reproducibility efforts and comparisons that include "obvious" baselines, matching the experiences from other areas of machine learning as well as earlier work by Shchur et al. (2018) on reproducing experimental results for GNNs on citation network tasks.

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

# A    HYPERPARAMETER SEARCH SPACES

## A.1    PPI

For all models, a full grid search considering all combinations of the following parameters was performed:

- hidden_size $\in \{192, 256, 320\}$ - size of per-node representations.
- graph_num_layers $\in \{2, 3, 4, 5\}$ - number of propagation steps / layers.
- graph_layer_input_dropout_keep_prob $\in \{0.8, 0.9, 1.0\}$ - dropout applied before propagation steps.

## A.2    QM9

For all models, 500 configurations were considered, sampling hyperparameter settings uniformly from the following options:

- hidden_size $\in \{64, 96, 128\}$ - size of per-node representations.
- graph_num_layers $\in \{4, 6, 8\}$ - number of propagation steps / layers.
- graph_layer_input_dropout_keep_prob $\in \{0.8, 0.9, 1.0\}$ - dropout applied before propagation steps.
- layer_norm $\in \{True, False\}$ - decided if layer norm is applied after each propagation step.
- dense_layers $\in \{1, 2, 32\}$ - insert a fully connected layer applied to node representations between every dense_layers propagation steps. (32 effectively turns this off)
- res_connection $\in \{1, 2, 32\}$ - insert a residual connection between every res_connection propagation steps. (32 effectively turns this off)
- graph_activation_function $\in \{relu, leaky\_relu, elu, gelu, tanh\}$ - non-linearity applied after message passing.
- optimizer $\in \{RMSProp, Adam\}$ - optimizer used (with TF 1.13.1 default parameters).
- lr $\in [0.0005, 0.001]$ - learning rate.
- cell $\in \{RNN, GRU, LSTM\}$ - gated cell used for GGNN (only part of search space for GGNN).
- num_heads $\in \{4, 8, 16\}$ - number of attention heads used for R-GAT (only part of search space for R-GAT).

## A.3    VARMISUSE

For all models, a full grid search considering all combinations of the following parameters was performed:

- hidden_size $\in \{64, 96, 128\}$ - size of per-node representations.
- graph_num_layers $\in \{6, 8, 10\}$ - number of propagation steps / layers.
- graph_layer_input_dropout_keep_prob $\in \{0.8, 0.9, 1.0\}$ - dropout applied before propagation steps.
- cell $\in \{GRU, LSTM\}$ - gated cell used for GGNN (only part of search space for GGNN).
- num_heads $\in \{4, 8\}$ - number of attention heads used for R-GAT (only part of search space for R-GAT).

