# OpenReview forum: "GNN-FiLM: Graph Neural Networks with Feature-wise Linear Modulation"
_ICLR.cc/2020/Conference — Reject_

### Official Review · AnonReviewer3 · 2019-10-16
**Official Blind Review #3**

**Rating:** 8

**Review:**

The paper proposes a new Graph Neural Network (GNN) architecture that uses Feature-wise Linear Modulation (FiLM) to condition the source-to-target node message-passing based on the target node representation. In this way, GNN-FiLM aims to allow a GNN's message propagation to "focus on feature that are especially relevant for the update of the target node." The authors clearly describe prior GNN architectures, showing that the do not incorporate such forms of message propagation. The authors then describe several intuitive ways of adding such a form of message propagation, before describing why those approaches do not work in practice. Finally, the authors introduce GNN-FiLM, which is computationally reasonable and works well in practice, as evaluated according to several GNN benchmarks. The GNN-FiLM model is also quite simple and elegant, which makes me think it is likely to work on more tasks than the authors experiment on.

The paper is clear and easy to follow. The authors' description of other GNNs architectures is clear, and their own approach seems well motivated and clearly described in relation to previous GNNs.

The authors have released the code for method, where they have also reimplemented several popular GNN methods. The open-source codebase also seems to be valuable contribution for future research and reproducibility in work on GNNs.

The empirical evaluation seems thorough. GNN-FiLM works well on 3 graph tasks (PPI, QM9, VarMisuse) with different properties. To compare models, the authors conduct a search of hyperparameter ranges for each model. The authors even improve several of the baseline methods, generalizing some approaches to include different edge types and to add self-loops, as well as using better networks (adding dropout and increasing hidden dimensions). The paper even finds that one existing/obvious GNN architecture (GNN-MLP) is underrated. The paper reads like an honest analysis of existing methods, even though it also introduces its own, new method that works better.

Questions:
* Do you have any intuition about why the Eqn. 5 model is less stable than GNN-FiLM?
* On GNN-FiLM's training stability and regularization: I am interested in the negative result described in the following sentence: "Preliminary experiments on the citation network data showed results that were at best comparable to the baseline methods, but changes of a random seed led to substantial fluctuations (mirroring the problems with evaluation on these tasks reported by Shchur et al. (2018))" Do the authors have any intuition about why the results are highly dependent on the random seed? The results on other tasks seems to not have much variance. It could be interesting to try various techniques to stabilize learning on that task. A simple approach like gradient clipping might work, or there are perhaps other techniques. For example, in "TADAM: Task dependent adaptive metric for improved few-shot learning", the authors find it important to regularize FiLM parameters \gamma and \beta towards 1 and 0, respectively, which may make learning more stable here. In general, previous work seems to find it important to regularize the parameters that predict FiLM parameters, which may also fix GNN-FiLM's overfitting on VarMisuse. Exploring such approaches could make it easier for future work to use FiLM with GNNs.
* On potential related work: GNN-FiLM is a "self-conditioned" model which learns to apply feature-wise transformations based on the activations at the current layer. If I am not mistaken, the self-conditioning aspect of GNN-FiLM makes it related to the self-conditioned models described in "Feature-wise Transformations" ([Feature-wise transformations](https://distill.pub/2018/feature-wise-transformations/)) - for example, see the "Image Recognition" section (Squeeze-and-Excitation Networks and Highway Networks) and the "Natural Language Processing" section (LSTMs, gated linear units, and gated-attention reader). Do the authors see a connection with such models? If so, it would be interesting to hear these works discussed (in the rebuttal and paper) and the exact connection described.

**Experience Assessment:**

I have published one or two papers in this area.

**Review Assessment: Checking Correctness Of Derivations And Theory:**

N/A

**Review Assessment: Checking Correctness Of Experiments:**

I assessed the sensibility of the experiments.

**Review Assessment: Thoroughness In Paper Reading:**

I read the paper thoroughly.

---

> ### Author Response · Authors · 2019-11-07
> **Discussion of Review #3**
>
> Thank you for your kind review and your interesting questions!
>
> > * Do you have any intuition about why the Eqn. 5 model is less stable
> > than GNN-FiLM?
>
> While there are no clean experiments to back this up, the instability seems to have two sources:
> (1) The computation of $\mathbf{W}_{\ell,t,v,c}$ takes a state $h_v$ of size $d$ and blows it up into a $d \times d$ matrix. Obviously, the values (or worse, outliers) of a single dimension of $h_v$ then affect a number of values in $\mathbf{W}_{\ell,t,v,c}$, which are then applied to some $h_u$. This blow-up-then-collapse pattern exacerbates the effect of outlier data. Using squashing activation functions such as sigmoids to limit the range of values helped with this, but seems to have not entirely solved the problem.
> (2) Computing $\mathbf{W}_{\ell,t,v,c}$ by a linear layer $W_f$ from $h_v$ requires careful initialisation of $W_f$ to ensure that $\mathbf{W}_{\ell,t,v,c}$ is appropriate (e.g., that the sum of each row is near 1, to avoid strong growth / decay in values). This somewhat worked, but some unlucky initializations would still diverge. Doing a deeper analysis and then apply, for example, appropriate rescaling of weights after init (maybe in the style of Fixup Initialization https://arxiv.org/pdf/1901.09321.pdf) could work, but was not explored as the FiLM variant worked well, and the memory and time cost of computing $\mathbf{W}_{\ell,t,v,c}$ is substantial.
>
> > * On GNN-FiLM's training stability and regularization: I am interested
> > in the negative result described in the following sentence:
> > "Preliminary experiments on the citation network data showed results
> > that were at best comparable to the baseline methods, but changes of a
> > random seed led to substantial fluctuations (mirroring the problems
> > with evaluation on these tasks reported by Shchur et al. (2018))" Do
> > the authors have any intuition about why the results are highly
> > dependent on the random seed? The results on other tasks seems to not
> > have much variance. It could be interesting to try various techniques
> > to stabilize learning on that task.
>
> This seems to be a property of the datasets, see the error bars of the plots in appendix D of Shchur et al. (2018) (https://arxiv.org/pdf/1811.05868.pdf).
>
> > A simple approach like gradient clipping might work, or there are perhaps
> > other techniques.
>
> Gradient clipping is implemented for all evaluated models (with a max grad norm of 1.0, though different clipping values had little influence on results in cursory experiments).
>
> > For example, in "TADAM: Task dependent adaptive metric for improved
> > few-shot learning", the authors find it important to regularize FiLM
> > parameters \gamma and \beta towards 1 and 0, respectively, which may
> > make learning more stable here. In general, previous work seems to
> > find it important to regularize the parameters that predict FiLM
> > parameters, which may also fix GNN-FiLM's overfitting on VarMisuse.
> > Exploring such approaches could make it easier for future work to use
> > FiLM with GNNs.
>
> Using a regularizer to explicitly push $\gamma$ to 1 and $\beta$ to 0 sounds like a reasonable idea, which has not been used in the experiments yet. If time allows, experiments with such a penalty term will be run during the rebuttal period.
>
> > * On potential related work: GNN-FiLM is a "self-conditioned" model
> > which learns to apply feature-wise transformations based on the
> > activations at the current layer. If I am not mistaken, the
> > self-conditioning aspect of GNN-FiLM makes it related to the
> > self-conditioned models described in "Feature-wise Transformations"
> > ([Feature-wise
> > transformations](https://distill.pub/2018/feature-wise-transformations/))
> > - for example, see the "Image Recognition" section
> > (Squeeze-and-Excitation Networks and Highway Networks) and the
> > "Natural Language Processing" section (LSTMs, gated linear units, and
> > gated-attention reader). Do the authors see a connection with such
> > models? If so, it would be interesting to hear these works discussed
> > (in the rebuttal and paper) and the exact connection described.
>
> Yes, GNN-FiLM clearly fits into this general (fairly large) class of self-conditioned works, though it is unclear what direct conclusions to draw from this. Self-conditioning is clearly a substantial trend at the moment (primarily in the form of attention mechanism), often used as step towards multi-step reasoning (in which results on layer $\ell$ are used to determine computation at layer $\ell+1$).
> While a general connection to feature-wise examples of this exists, it does not seem clear how to exploit this for deeper understanding or better results at this time.

---

> > ### Comment · AnonReviewer3 · 2019-11-12
> > **Response for Review #3**
> >
> > Thank you for your response - your answers are reasonable to me.
> >
> > Regarding the other reviewers' remarks:
> > - I do agree with R2 that it would be nice to explicitly cite more work (for example, to the self-conditioned models I mentioned in my review), even if the connection does not immediately yield some deeper understanding (just a general intuition).
> > - I also agree with R4/R2 that a visualization of the method would be useful.
> > - R4 commented that the story of the paper is confusing with both strong baselines and a new method. I am inclined to agree with the authors that this point isn't a con (and is probably a pro), even if the story isn't as straightforward. Because the baselines are so strong, I really feel that I can trust the authors on their results that GNN-FiLM is better (even though the margin may be limited). That makes this paper more valuable a contribution than existing work on GNNs (in my opinion).

---

### Official Review · AnonReviewer2 · 2019-10-21
**Official Blind Review #2**

**Rating:** 3

**Review:**

This paper performs a concise experimental survey of graph neural network method, and introduce GNN-FiLM as a new approach. They reproduce all the baseline and show that GNN-FiLM is in line with SOTA models.

The paper is pretty easy to follow, and the intuition is clearly explained. The idea is pretty simple, and it is a natural extension of FiLM to the graph setting. The results are tested on 4 different datasets, and the experimental protocol is clearly explained, and the results are convincing.)

However, I am missing a discussion of the proposed methods. For instance, what are the FiLM clusters? What are the key errors with GNN-FiLM, GNN-MLP0? Are the models complementary (e.g., by using a mixture of experts)? Can we combine GNN-MLP0 and GNN-FiLM?

Remarks:
 - The bibliography work is incomplete. For instance, the authors do not cite the Hypernetwork paper [0], when stating that several GNN variant exists, we expect to have some references, idem when the authors mention that [FiLM] has been very effective in several domain (GAN [1], Meta-learning [2] Sound [3], language-vision task [4]). As a hint, it is pretty rare to have empirical papers with only 16 citations. I would say that 20-25 is the bare minimum.
 - Adding a visual sketch of the different models may provide additional intuition
 - I would encourage the author to define the dimension of W_l, h_u, etc. clearly.
 - On a personal note, I like informal writing and paper honesty, but I would not always recommend it for submission.
 - As the authors reproduce the experiments, It would have been useful to add the original results in the table whenever it is possible
 - The paper gives a feeling that it lacks some rigor (missing ref, detected bug, mathematical formalism is light); it makes me a bit skeptic regarding some experimental conclusion. It would have been helpful to look at the code (at least to see if it can be easily parsed and analyzed.)

I think the paper is valuable for the community as it provides a simple survey of the current methods, the code is available, and the authors provide a lot of intuition. However, the overall level is slightly below the ICLR level for the following reason: incomplete bibliography, lack of complementary experiments, lack of discussion. The overall writing style is quite unusual, but It is not a negative point from my perspective. In its current state, it is a strong workshop submission but a not-good-enough paper for ICLR.

However, I am open to discussion, especially if the authors have complementary discussions and results.

PS: the reviewer is very familiar with the modulation and multi-modal literature, but he has only basic knowledge in graph networks.


[0] Ha, David, Andrew Dai, and Quoc V. Le. ICLR 2017.
[1] Brock, Andrew, Jeff Donahue, and Karen Simonyan. "Large scale gan training for high fidelity natural image synthesis." ICLR 2019
[3] Jiang, X., Havaei, M., Varno, F., Chartrand, G., Chapados, N., & Matwin, S.. Learning to learn with conditional class dependencies. ICLR 2019
[3] Abdelnour, Jerome, Giampiero Salvi, and Jean Rouat. "From Visual to Acoustic Question Answering." arXiv preprint arXiv:1902.11280 (2019).
[4] Strub, F., Seurin, M., Perez, E., De Vries, H., Mary, J., Preux, P., & CourvilleOlivier Pietquin, A.. Visual reasoning with multi-hop feature modulation. ECCV 2018

**Experience Assessment:**

I have published one or two papers in this area.

**Review Assessment: Checking Correctness Of Derivations And Theory:**

N/A

**Review Assessment: Checking Correctness Of Experiments:**

I assessed the sensibility of the experiments.

**Review Assessment: Thoroughness In Paper Reading:**

I read the paper at least twice and used my best judgement in assessing the paper.

---

> ### Author Response · Authors · 2019-11-06
> **Discussion of Review #2 (Part 1/2)**
>
> First, thank you for your careful review of this submission. Note that this reply is split into two parts for character limit reasons.
>
> > This paper performs a concise experimental survey of graph neural
> > network method, and introduce GNN-FiLM as a new approach. They
> > reproduce all the baseline and show that GNN-FiLM is in line with SOTA
> > models.
>
> Please note that the generalisations to the relational setting for the GAT (and now GIN) baselines are also novel contributions of this paper.
>
> > The paper is pretty easy to follow, and the intuition is clearly
> > explained. The idea is pretty simple, and it is a natural extension of
> > FiLM to the graph setting. The results are tested on 4 different
> > datasets, and the experimental protocol is clearly explained, and the
> > results are convincing.)
>
> Note: The models are only properly evaluated on three datasets (the Citation Network datasets are not properly considered in the paper).
>
> > However, I am missing a discussion of the proposed methods. For
> > instance, what are the FiLM clusters? What are the key errors with
> > GNN-FiLM, GNN-MLP0? Are the models complementary (e.g., by using a
> > mixture of experts)? Can we combine GNN-MLP0 and GNN-FiLM?
>
> The requested error analysis of the different models is desirable, but it is not clear how to achieve it. The datasets themselves are not annotated with additional meaningful labels that could be used for deeper analysis (e.g., "VarMisuse problems of type A show worse performances"), and hence only a qualitative analysis of the graphs would be possible. However, these graphs have thousands of nodes, so its unclear how to present such an analysis in a paper.
>
> What _can_ be done is to quantitively study if the models are complimentary in a mixture-of-experts setting, and we will try to provide numbers for this (for VarMisuse, where the semantics of a mixture of experts model are most clearly defined and room for improvements exists).
>
> Finally, it would be nice if you could clarify what you mean by "FiLM clusters." Some papers reporting on the use of FiLM do display PCA visualisations of embeddings of either the actual \alpha/\beta values or the embeddings before/after application of the FiLM layer. If you are asking for such visualisations here, it is unclear what they would look like, as the FiLM-conditioning is based on the embeddings of other graph nodes. Again, as graphs (and nodes) are not provided with a small set of interpretable labels, it's unclear what information cluster visualisations of the embeddings would provide.

---

> > ### Author Response · Authors · 2019-11-08
> > **Analysis of differences between model results**
> >
> > To probe the question of the respective strengths of different models, the VarMisuse experiments were extended for the four best-performing architectures (RGCN, RGIN, GNN-MLP1, GNN-FiLM).
> >
> > Question 1: What is the overlap in results? For this, the groups of five models (to minimise the differences between different random seeds) trained for each architecture were compared on the SeenProjTest dataset, measuring how many examples were predicted correctly by any of the models from one architecture, but not by the other. Substantial difference here would indicate that there are large classes of examples that are only solved by one architecture, but not by another.
> > RGCN vs RGIN: 2.2% could only be solved by RGCN models, 2.7% could only be solved by RGIN models.
> > RGCN vs GNN-MLP1: 2.3% only by RGCN, 2.8% only by GNN-MLP1
> > RGCN vs GNN-FiLM: 2.5% only by RGCN, 2.8% only by GNN-FiLM
> > RGIN vs GNN-MLP1: 2.0% only by RGIN, 2.0% only by GNN-MLP1
> > RGIN vs GNN-FiLM: 2.1% only by RGIN, 2.0% only by GNN-FiLM
> > GNN-MLP1 vs GNN-FiLM: 2.1% only by GNN-MLP1, 2.0% only by GNN-FiLM
> >
> > This seems to indicate that there are no substantial differences between the kinds of examples solved by different models.
> >
> > Question 2: Does ensembling different model types boost results? For this, groups of models per architecture were ensembled by averaging their (classification) outputs before computing the accuracy on the SeenProjTest dataset:
> > 5 RGCN models:  89.3% accuracy
> > 5 RGIN models: 89.2% accuracy
> > 5 GNN-MLP1 models: 89.1% accuracy
> > 5 GNN-FiLM models: 89.0% accuracy
> > 4 RGCN + 1 GNN-FiLM: 89.7% accuracy
> > 3 RGCN + 2 GNN-FiLM: 90.0% accuracy
> > 2 RGCN + 3 GNN-FiLM: 90.1% accuracy
> > 1 RGCN + 4 GNN-FiLM: 89.6% accuracy
> >
> > This seems to indicate that the models do have different biases, and mixing models with different biases is an effective way of improving results.
> >
> > Overall, the results seem to be in line with general results on ensembling of models: Ensembling improves results; ensembling models with slightly different architectures improves further. However, the models are not fully complimentary.

---

> > > ### Comment · AnonReviewer2 · 2019-11-15
> > > **Response**
> > >
> > > Thank you very much for your responses and your additional experiments.
> > >
> > > I liked that you added results from the literature as it demonstrates the quality of the experiments you had (i.e., you improved other people baseline). The ensemble methods are well-done, and they are valuable assets.
> > >
> > > I also want to remove the ambiguity here:
> > > "This part of the review is quite puzzling. What does "detected bug" refer to? Where is mathematical formalism light, i.e., are there parts of the model that are undefined, or definitions that are imprecise? As discussed above, dimensionality choices were intentionally omitted as they are hyperparameters that are varied in practice."
> > >
> > >  * "detected bug" -> "a small implementation bug brought focus to the" page 5 ;)
> > >  * There exist different mathematical formalism to deal with various size dimensions, e.g., a dot, or mentioning that the X dimension can be extended to multiple dimension, for instance.
> > >
> > > Regarding the citation, it is important to put a paper in its context, and to always support a claim with previous works. For instance,  when saying that an approach (modulation) has been applied to several topics, one needs to provide examples. In this case, I helped you by providing several citations from various authors, topics, and laboratories to support my point. More generally, I believe that even "trivial" citations such as LSTM, hypernetworks etc. are important; it never hurts to cite people and to go outside the scope of pure "useful" citations.
> > >
> > > I acknowledge the author's efforts toward improving paper quality with more experiments. I really do! However, I think that the paper may still need further reflection to improve the overall message, e.g., provide insightful sketches, increase maths clarity, better disambiguate the paper contributions vs. reproducibility parts, etc.
> > >
> > > I also encourage the authors to think about exploring analysis tools. For instance, t-sne of FiLM parameters may highlight some behavior of the methods, and outliers may indicate which cases are not correctly handled. I recommend the FiLM distill articles where the authors took quite some effort to analyze the modulation properties with different experiments: https://distill.pub/2018/feature-wise-transformations/
> > > In other words, I think that there is room for more analyzing the proposed method that would complement the newly reported scores. (t-sne, error analysis, ablation etc.).
> > >
> > > Even if I find the experimental setting very solid, I think that this paper still needs to be matured before being published.

---

> > > > ### Author Response · Authors · 2019-11-15
> > > > **Discussion of Review#2**
> > > >
> > > > Thank you for your time. Would you be able to point out which change you believe would be most significant to improve the paper? Conversely, given that it is now almost a page over the page limit, what content do you believe should be omitted to make space for the new content you are proposing?

---

> ### Author Response · Authors · 2019-11-06
> **Discussion of Review #2 (Part 2/2)**
>
> (This is part 2 of the author reply)
>
>
> > Remarks:
> >  - The bibliography work is incomplete. For instance, the authors do
> >  not cite the Hypernetwork paper [0], when stating that several GNN
> >  variant exists, we expect to have some references, idem when the
> >  authors mention that [FiLM] has been very effective in several domain
> >  (GAN [1], Meta-learning [2] Sound [3], language-vision task [4]). As
> >  a hint, it is pretty rare to have empirical papers with only 16
> >  citations. I would say that 20-25 is the bare minimum.
>
> The reference to the Hypernetwork paper was indeed missing (now included), but overall, references in the paper are intentionally restricted to works consulted in the preparation of the manuscript, and especially focus on those papers that are crucial to the understanding of the used techniques (e.g., the paper introducing FiLM layers) or to provide further pointers for methods compared against (the references to compared-to GNNs), or to papers providing context for the tasks used for evaluation.
> If you believe that citing more papers would be helpful, it would be helpful if you could explain what value you expect these references to provide to the reader of this paper.
>
> >  - Adding a visual sketch of the different models may provide
> >    additional intuition
>
> Several such sketches were attempted in the preparation of this paper, but it seemed impossible to find a format in which the differences between interactions between different node representations become clear. An idea for such a visualisation would be very much appreciated.
>
> >  - I would encourage the author to define the dimension of W_l, h_u,
> >    etc. clearly.
>
> These have been left out intentionally, as these dimensions are hyperparameter choices, which are indeed picked different in the experiments. Hence, these dimensions would remain variables that are only constrained by "dimensions of operations need to match", which does not seem to add substantial value.
>
> As example, consider the case of Eq. (2). Here, the shape of $h_v^{(t)}$ could be defined as $d^{(t)}$, and the shape of $W_\ell$ as $d^{(t+1)} \times d^{(t)}$. If one desires to apply the same layer for several timesteps, then $d^{(t+1)} = d^{(t)}$, otherwise the two can be chosen freely.
> This explanation does not seem to add substantial value, but creates relatively high cognitive load, and hence was omitted from the paper.
>
> >  - As the authors reproduce the experiments, It would have been useful
> >    to add the original results in the table whenever it is possible
>
> Tables 1 and 3 now include this information.
>
> >  - The paper gives a feeling that it lacks some rigor (missing ref,
> >  detected bug, mathematical formalism is light); it makes me a bit
> >  skeptic regarding some experimental conclusion. It would have been
> >  helpful to look at the code (at least to see if it can be easily
> >  parsed and analyzed.)
>
> This part of the review is quite puzzling. What does "detected bug" refer to? Where is mathematical formalism light, i.e., are there parts of the model that are undefined, or definitions that are imprecise? As discussed above, dimensionality choices were intentionally omitted as they are hyperparameters that are varied in practice.
>
> A tarball of a quickly anonymised version of the source code is now linked in a private comment (i.e., grep does not find author names or institution anymore) to allow to judge the code.
>
> > I think the paper is valuable for the community as it provides a
> > simple survey of the current methods,
>
> This is an incomplete summarisation of the paper: This is not a survey paper, but introduces new technical content. Concretely, its contributions are:
>  (1) A new GNN type based on the integration of the FiLM idea.
>  (2) Generalisation of existing GNN types (GAT, and now GIN) to the multi-relational setting.
>  (3) A substantial evaluation of (1) and (2) on a variety of tasks, with results that are showing that the new models from (1) and (2) outperform the state of the art.
>  (4) The inclusion of usually omitted baselines (GNN-MLP) which are also outperforming the state of the art and are competitive with the newly introduced models. This pointing to a larger issue in the current empirical evaluations of GNNs, and in hindsight, seems to be the most important contribution of this paper.

---

### Official Review · AnonReviewer4 · 2019-11-06
**Official Blind Review #4**

**Rating:** 6

**Review:**

This paper introduces a new type of Graph Neural Network (GNN) that incorporates Feature-wise Linear Modulation (FiLM) layers. Current GNNs update the target representations by aggregating information from neighbouring nodes without taking into the account the target node representation. As graph networks might benefit from such target-source interactions, the current work proposes to use FiLM layers to let the target node modulate the source node representations. The authors thoroughly evaluate this new architecture — called GNN-FiLM—-on several graph benchmarks, including Citeseer, PPI, QM9, and VarMisuse. The proposed network outperforms the other methods on QM9 and is on par on the other benchmarks.

Strengths
- The literature review of existing work on GNN was a pleasure to read and provided a good motivation for the proposed GNN-FiLM architecture.
- The authors put significant effort into reproducing other GNN baselines. Perhaps the most surprising result of this work is that all GNNs perform remarkably similar (contrary to what previous work has reported)

Weaknesses
- There seems to be a much tighter relationship between GNN-FiLM and Gated GNNs than currently discussed. If you actually write down the equations of the recurrent cell $r$ in Eq 1), you’ll notice that there are feature-wise interactions between the target node representations and the (sum of) source node representations. The paper should discuss in more depth what the exact differences are. Some visualizations would also help here.
- Related to the previous point, I’d like to see a bit more discussion on *why* tight interactions between target and source nodes are helpful. For example, one could perhaps provide a toy example for which that’s obviously the case.

All in all, I believe the ideas and results of this paper are promising but insufficient for publication in its current form. The paper tries to communicate two messages: 1) a model paper arguing for GNN-FiLM, 2) an unbiased evaluation of existing GNN models, showing that their performance is surprisingly similar on a number of benchmarks (with equal hyperparameter search). Both points are interesting but are not worked out sufficiently to pass the bar. For 1), I’d like to see an in-depth discussion on the benefits of target-source interactions, providing more insights into why this might be beneficial. Your current experiments report very minimal gains for your proposed network, questioning why such interactions might be necessary in the first place. For 2), I’d suggest to rewrite the paper from a slightly different angle and add more graph benchmarks if available (disclaimer: I’m not in the graph network community, so I can’t fully evaluate how significant these results are)


Typos
——-
Last paragraph of intro: “two two” - > two

*EDIT
After reading the rebuttal and revised paper, I've updated my score to "Weak Accept". The toy example and connection to GGNN are important additions which makes the paper more complete, though I believe there's room for further improvement here (see comments below). All in all, I weakly recommend to accept the paper.

**Experience Assessment:**

I have read many papers in this area.

**Review Assessment: Checking Correctness Of Derivations And Theory:**

N/A

**Review Assessment: Checking Correctness Of Experiments:**

I assessed the sensibility of the experiments.

**Review Assessment: Thoroughness In Paper Reading:**

I read the paper at least twice and used my best judgement in assessing the paper.

---

> ### Author Response · Authors · 2019-11-07
> **Discussion of Review #4**
>
> Thank you for your kind (emergency?) review and engaging with this submission.
>
> > - There seems to be a much tighter relationship between GNN-FiLM and
> >   Gated GNNs than currently discussed. If you actually write down the
> >   equations of the recurrent cell  in Eq 1), you’ll notice that there
> >   are feature-wise interactions between the target node representations
> >   and the (sum of) source node representations.
>
> This is indeed true (for GRU/LSTM cells) and should maybe discussed in the paper in more depth, though it is a bit unclear where to best do this to not interrupt the flow of presentation.
> The differences arise from the application of the gated cell after summation of incoming messages. Concretely, the "forgetting" of memories in a GRU/LSTM is similar to modulation of messages from the self-loop edges, and the gating of the cell input is similar to the modulation of incoming messages. However, as GNN-FiLM uses _different_ $\gamma_\ell$/$\beta_\ell$ values for different edge types $\ell$, the modulation is additionally dependent on the kind of relationship between nodes. In the case of a single edge type (+ a fresh self-loop edge type), GGNNs are indeed mathematically very similar to GNN-FiLM.
>
> > - Related to the previous point, I’d like to see a bit more discussion
> >   on *why* tight interactions between target and source nodes are
> >   helpful. For example, one could perhaps provide a toy example for
> >   which that’s obviously the case.
>
> A simple toy example may be the following: Assume nodes are of type VA and type VB, and there are two edge types E1 and E2. Our task is to count the number of E1-neighbours of VA nodes, and the number of E2-neighbours of VB nodes. In FiLM, we can "filter" edges by using $\gamma_{E1} = 1, \gamma_{E2} = 0$ for VA nodes and $\gamma_{E1} = 0, \gamma_{E2}=1$ for VB nodes. Hence, GNN-FiLM can solve this task using a single layer. Other models obviously can solve this task as well with more layers and feature dimensions by counting VA-E1, VA-E2, VB-E1, VB-E1 neighbours separately in a first message passing step, and then projecting to the desired dimension in a second layer.
>
> More generally, as the modulation of messages depends on the edge target representation and the edge type, GNN-FiLM can learn to (softly) select a subset of the graph edges for message passing, conditioned on the current representation of nodes, _per feature dimension_ (this is a substantial difference to the newly-introduced R-GAT setting; the original GAT model has no notion of edge types). For example, in the Program Graph setting of the VarMisuse task, it may choose to emphasise "NextVariableUse" edges for some dimensions to gain information about how a variable is used later.
>
> You're right that this intuition is not provided clearly in the paper at the moment, and will be added in the next revision (but maybe only after feedback if these explanations make sense to a reader).
>
> > Your current experiments report very minimal gains for your proposed
> > network, questioning why such interactions might be necessary in the
> > first place.
>
> Note that the experimental gains over _standard_ baselines are quite substantial:
> * For Tab. 1, the reported state of the art is a Micro-F1 of 0.973 +/- 0.002
> * For Tab. 2, there is no clear state of the art as many papers use slightly different experiment design for this task (e.g., provide more features, ...); but the standard models are GGNN/R-GCN.
> * For Tab. 3, the reported state of the art is 84.0% (on SeenProjTest) and 74.1% (on UnseenProjTest) accuracy.
> While more graph tasks are studied, these 3 provide a good coverage of graph tasks (from small to large graph, classification and regression, node-level and graph-level). The trends in these results are very likely to hold for the majority of tasks.
>
> Early feedback (and curiosity) led to more and more baselines in the experiments, and existing baselines were improved in a number of ways (e.g., the generalisations to the multi-relational setting, which are crucial for the performance in the reported experiments).
>
> Your review fits exactly to fear around submitting this paper: By doing a very thorough job on the experiments, the importance of the new method looks smaller; doing a worse job on the experiments (not including GNN-MLP*; not extending GAT/GIN to R-GAT/R-GIN) would have made it look better. These nuances cannot be obvious to someone who's not active in the GNN research community (and so this rant is not meant as a "you did a bad job as a reviewer"), but this explanation may help to understand the context.
>
> However, the result that the differences between well-tuned, equally-well implemented models are small (and that the GNN-MLP baselines outperform other models) seems to be scientifically quite relevant, and of particular importance to the GNN research community. Disseminating this result more widely should be a reason for acceptance for publication.

---

> > ### Comment · AnonReviewer4 · 2019-11-07
> > **Thank you for the clarification**
> >
> > Please incorporate this response (in-depth discussion of relation to GGGN, toy example) in your next revision and I'd willing to update my scores accordingly.
> >
> > As for your last point, thanks for providing some more context. I'm not in the GNN research community so I can't really assess the importance of your results for this community. I'll leave this to the other reviewers and AC.

---

> > > ### Author Response · Authors · 2019-11-07
> > > **Discussion of Review #4**
> > >
> > > A new revision of the paper was just uploaded, including the GGNN relation bits / toy example in Sect. 2.1 (under Eq. (7)). Again, thank you for engaging with this submission, even though it is outside of your immediate research interests.

---

> > > ### Author Response · Authors · 2019-11-13
> > > **Discussion of Review #4**
> > >
> > > This is always awkward, but anyway: You said you would be willing to update your score if the paper is  updated according to your suggestions; did you have time to check the changes to see if you are satisfied with them, and could you update the score if you are (or indicate what's missing in your eyes)?
> > >
> > > Thank you.

---

> > > > ### Comment · AnonReviewer4 · 2019-11-15
> > > > **Updated my score**
> > > >
> > > > Sorry for the late response. I've read your revised submission and will update to a weak accept.
> > > >
> > > > There are a few more changes that could improve the completeness of this manuscript: 1) write down the actual equations for GNN-FILM and GGNN and point out the differences (the current explanation is too high-level). You could very easily move this into the appendix. 2) I'm still missing a visualization of the proposed model.

---

### Author Response · Authors · 2019-11-05
**Submission revision 2: summary of changes**

At the beginning of the rebuttal period, the submission was updated with new results obtained after the submission deadline (i.e., _not_ taking the reviews into account). Concretely, two substantial changes were made:
(a) In Sect. 3.3.3, the VarMisuse results were updated to reflect five runs per model, providing error bars.
(b) Throughout the paper, Graph Isomorphism Networks (GINs) were added as an additional baseline. This means that they now get introduced in Sect. 2, where they are also generalised to the relational setting (now Eq. (4); and they are evaluated in Sect. 3 for all tasks, roughly reaching results between R-GAT and the GNN-FiLM/MLP models.

---

> ### Author Response · Authors · 2019-11-07
> **Submission revision 3: summary of changes**
>
> In response to comments from the reviewers, the submission was updated to include the following changes:
> * Cited Ha et al. (2017) for their HyperNetwork paper.
> * Included a toy example illustrating the usefulness of FiLM layers.
> * Included a note on the relation of GGNN and GNN-FiLM.
> * Included baseline results from the literature in experimental tables.

---

### Public Comment · ~Uri_Alon1 · 2019-12-30
**Really nice code repository**

For future reference -
Although this paper is unfortunately rejected, I would like to highlight this paper's quality code repository at https://github.com/Microsoft/tf-gnn-samples .

The repository implements seven GNN architectures, and each of them can be trained and tested on four tasks including popular benchmarks.
As far as I tried, the paper seems to be fully reproducible using the repository.
It also seems to be easy to experiment with novel GNN architectures by simply extending and modifying the existing classes.
This is a great example of how an ML paper's code repository should be.

Thank you for sharing the code, I am sure that it would facilitate future research in this area.

---

### Decision · Program_Chairs · 2019-12-19

**Decision:**

Reject

**Comment:**

While considerable effort went into improving the paper during the author response period, the concerns outlined by reviewer 2 remain and the aggregate score across reviewers reflects this issue. The AC recommends rejection with strong encouragement to resubmit this work to another high quality venue upon further revision to the work.